# *Wuschel2* enables highly efficient CRISPR/ Cas-targeted genome editing during rapid *de novo* shoot regeneration in sorghum

Ping Che [1✉], Emily Wu[1], Marissa K. Simon[1], Ajith Anand[1], Keith Lowe[1], Huirong Gao [1], Amy L. Sigmund[1], Meizhu Yang[1], Marc C. Albertsen [1], William Gordon-Kamm[1] & Todd J. Jones [1]

For many important crops including sorghum, use of CRISPR/Cas technology is limited not only by the delivery of the gene-modification components into a plant cell, but also by the ability to regenerate a fertile plant from the engineered cell through tissue culture. Here, we report that *Wuschel2* (*Wus2*)-enabled transformation increases not only the transformation efficiency, but also the CRISPR/Cas-targeted genome editing frequency in sorghum (*Sorghum bicolor L.*). Using *Agrobacterium*-mediated transformation, we have demonstrated *Wus2*-induced direct somatic embryo formation and regeneration, bypassing genotype-dependent callus formation and significantly shortening the tissue culture cycle time. This method also increased the regeneration capacity that resulted in higher transformation efficiency across different sorghum varieties. Subsequently, advanced excision systems and "altruistic" transformation technology have been developed to generate high-quality morphogenic gene-free and/or selectable marker-free sorghum events. Finally, we demonstrate up to 6.8-fold increase in CRISPR/Cas9-mediated gene dropout frequency using *Wus2*-enabled transformation, compared to without *Wus2*, across various targeted loci in different sorghum genotypes.

[1] Corteva Agriscience, Johnston, IA 50131, USA. ✉email: ping.che@corteva.com

The standard paradigm for generating CRISPR/Cas-mediated genome-edited crops currently combines the delivery of the gene-editing components into a plant cell either through Agrobacterium-mediated gene delivery or biolistic particle delivery, with regeneration of a plant from that cell through tissue culture[1,2]. After gene delivery, the conventional tissue culture process includes callus induction, callus proliferation, maturation, and rooting phases, in which the callus proliferation phase is typically the time-intensive and rate-limiting step because of profound genotype dependency[2–4]. Therefore, developing a more efficient transformation system that overcomes genotype-dependent barriers is critical for expanding the application of CRISPR/Cas-mediated genome-modification across a broad range of genotypes and applications[2].

Rapid somatic embryo formation from immature maize (Zea mays L.) scutella can be induced by regulating expression of the morphogenic genes Zm-Wuschel2 (Zm-Wus2) and Zm-Baby boom (Zm-Bbm), for direct regeneration from somatic embryos without an intervening callus phase, a method that is extending the useful genotype range for maize transformation[5,6]. Although both Wus2 and Bbm are essential for somatic embryo formation and shoot regeneration[6,7], Wus2, a bifunctional homeodomain transcription factor[8–10] and essential factor for de novo establishment of the shoot stem cell niche[8], plays a pivotal role in inducing the direct somatic embryo formation in tissue culture[5,6]. In addition to working synergistically with Bbm, it was recently reported that a strong pulse of Wus2 expression driven by the maize Pltp promoter was sufficient to rapidly stimulate somatic embryos in maize[3,6]. However, constitutive and strong expression of these morphogenic genes could cause undesired pleiotropic effects in regenerated plants, including reduced fertility[3,6]. To maximize the positive stimulation of somatic embryogenesis of morphogenic genes during tissue culture, but to obviate their adverse pleiotropic effects in T0 plants, several gene regulation systems have been developed that control morphogenic gene expression through either inducible expression, developmentally regulated expression, or excision[6]. Moreover, taking advantage of the attribute that WUS2 protein migrates into neighboring cells[11], a novel transformation system known as "altruistic" transformation that was developed to induce direct somatic embryo formation without Wus2 gene integration[12].

In this study, we recapitulated and extended those technologies to sorghum (Sorghum bicolor L.) and demonstrated efficient, genotype-independent transformation for generating high-quality morphogenic gene-free and/or selectable marker-free sorghum events. We then investigated CRISPR/Cas-mediated genome editing frequency using Wus2-enabled transformation and compared that to conventional transformation without Wus2 in sorghum. In all cases, we found that the mutation frequency (using a single guide) and gene-dropout frequency (using a pair of guides) using Wus2-enabled transformation was significantly higher than that of conventional transformation across various targeting loci in different sorghum genotypes.

## Results and discussion

**Overcoming genotype dependence using integrating morphogenic genes.** The efficiency of Agrobacterium-mediated transformation technology through callus induction, proliferation, and then maturation (hereby referred to as "conventional transformation", to distinguish it from the morphogenic gene- or Wus2-enabled transformation systems described below) is influenced by many factors such as Agrobacterium strains, VIR-containing accessory plasmids, infection, and cultivation regimes, and therefore, it is highly genotype-dependent[2–4,13]. Of those factors, it is believed that Agrobacterium-mediated gene delivery and

shoot regeneration through callus are the major obstacles for achieving genotype-independent transformation[2,3,6]. In this study, we used thymidine auxotrophic Agrobacterium strain (LBA4404 Thy-) harboring the ternary vector system with accessory plasmid pPHP71539[14,15] as a consistent underlying system in all experiments. Using this Agrobacterium strain containing the accessory plasmid to establish our baseline, we conducted experiments to dissect the main causes of genotype dependency for conventional sorghum transformation, starting by comparing the T-DNA delivery efficiency and the in vitro tissue culture response among three genotypes, Tx430, Tx623, and Tx2752. We purposely chose those genotypes for comparison because Tx430 is well-known as the most transformable sorghum genotype with an average transformation efficiency of around 20%, based on previously reported conventional transformation data[14], but Tx623 and Tx2752 are among the most recalcitrant sorghum genotypes and no successful transformation in these two genotypes has been reported. To evaluate T-DNA delivery efficiency, we performed a transient expression assay using previously reported binary vector pPHP45981 containing Zs-YELLOW1 as the visual marker[14]. As shown in Fig. 1a–c, roughly similar gene delivery was observed for all three genotypes based on the number of yellow fluorescent foci on the adaxial surfaces of the immature scutella, three days post-inoculation on co-cultivation medium (Supplementary Table 2). This suggests the ternary transformation system used in this study may support broad genotype-independent T-DNA delivery for sorghum, or at a minimum greatly extend the genotype range. Subsequently, tissue culture responses were evaluated two weeks after Agrobacterium infection on a multi-purpose medium (Supplementary Table 3) to induce callus proliferation without selection (Fig. 1d–f). As shown in Fig. 1d, prolific growth produced embryogenic callus on all surfaces of the immature scutella in Tx430. In contrast, tissue necrosis with almost no embryogenic growth was observed in genotypes Tx623 nor Tx2752 (Fig. 1e, f). Consequently, both Tx623 and Tx2752 were considered virtually non-transformable through conventional callus induction and proliferation procedures. Based on those observations, we concluded that for conventional Agrobacterium-mediated transformation of immature embryos, transient T-DNA delivery was efficient across the genotypes, but the tissue culture response to produce embryogenic callus was genotype-dependent, and therefore presented the primary bottleneck for achieving genotype-independent sorghum transformation.

Recently, it was reported that rapid somatic embryo formation from immature scutella can be induced by regulating morphogenic gene expression, such as Wus2 and/or Bbm, for direct shoot regeneration without an intervening callus phase in maize[3,5,6]. To evaluate this transformation technology in sorghum, we first characterized rapid somatic embryo formation in sorghum transformed with binary vector pPHP79066[3], previously tested in maize, carrying both Wus2 and Bbm ($Axig1_{pro}$:Wus2/$Pltp_{pro}$:Bbm) as morphogenic genes, Zs-YELLOW1 ($Ltp2_{pro}$:Zs-YELLOW1) as the visual marker and Hra ($Sb$-$Als_{pro}$:Hra) as a selectable marker[3] (Supplementary Fig. 1a). Transformation using these Wus2/Bbm expression cassettes was conducted by modifying the protocol published by Wu, et al.[16] without the nine-week prolonged selective callus proliferation phase (this truncated protocol being referred to as the morphogenic gene-enabled transformation system) (Supplementary Fig. 2). After Agrobacterium infection (Supplementary Table 1), the immature embryos were first cultured on co-cultivation medium (Supplementary Table 2) at 25 °C overnight followed by 28 °C for six more days in the darkness. The immature embryos were then subcultured on multi-purpose medium (Supplementary Table 3) without selection for another week to induce somatic embryo formation (Fig. 1g–i).

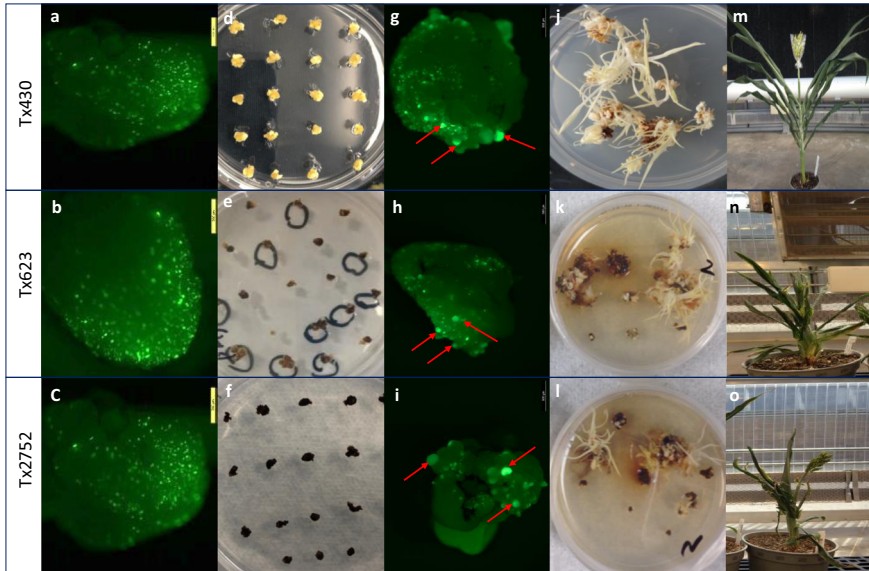

**Fig. 1 Characterization of sorghum transformation. a–c** Gene delivery capacity of ternary transformation system. Gene delivery was represented by the number of transgenic cells exhibiting YFP fluorescence on the surface of sorghum embryos. **d–f** Tissue culture response represented by callus proliferation after two weeks of *Agrobacterium* infection on the multi-purpose medium without selection. **g–i** YFP images of early-stage somatic embryo formation (indicated by the arrows) induced by morphogenic gene expression on the surface of immature scutella. **j–l** Shoot regeneration after four-week of maturation. **m–o** The pleiotropic impacts of morphogenic gene expression on single-copy QEs. **a**, **d**, **g**, **j** and **m** Tx430. **b**, **e**, **h**, **k**, and **n** Tx623. **c**, **f**, **i**, **l** and **o** Tx2752.

Similar to observations in maize[3], rapid somatic embryo formation was witnessed by 14 days after *Agrobacterium* infection for all three genotypes as indicated by the individual globular-shaped protrusions from the surface of immature scutella (Fig. 1g–i). Those somatic embryos were then transferred to maturation medium (Supplementary Table 5) with selection for four weeks to induce germination (Fig. 1j–l). After completing maturation, the germinated plantlets were transferred onto the rooting medium (Supplementary Table 6) for root development for one to three weeks. The total time from inoculation of immature embryos to transplantation of a fully developed transgenic plantlet to the greenhouse generally required only two months compared to up to four months for conventional transformation through callus induction. The overall transformation efficiency (calculated as the number of embryo with regenerated shoots per 100 embryos infected) with binary vector pPHP79066 was 38.8% for Tx430, almost double the efficiency of conventional transformation through callus formation (~20%) as previously reported[14] and as demonstrated using pPHP86655 (21.9%) in this study (Supplementary Fig. 1h and Table 1). The transformation efficiencies of Tx623 and Tx2752, virtually non-transformable through conventional callus induction and proliferation, were substantially improved to 6.5 and 9.5%, respectively (Table 1). These results demonstrate that the use of morphogenic gene expression obviates the need for genotype-dependent callus culture, substantially increasing transformation efficiency while simultaneously reducing the timeframe from *Agrobacterium* infection to the production of T0 plants. Significantly, this method also appears to overcome the genotype-dependent transformation barrier for previously non-transformable germplasms such as Tx623 and Tx2752, suggesting genotype-independent sorghum transformation. To further support this notion, in a separate set of experiments, six Corteva elite inbred lines were successfully transformed using vector pPHP94543 (*Axig1_{pro}:Wus2/Pltp_{pro}:Bbm*), a Corteva production construct with a proprietary trait gene, utilizing *NPTII* as a selectable marker, to achieve successful transformation in all six varieties, with efficiencies ranging from 1.4 to 24.7% (Supplementary Table 7).

Pleiotropic effects have been reported in transgenic maize plants carrying *Bbm* and *Wus2* using integrated morphogenic gene-enabled transformation[3,6]. To evaluate the impact of morphogenic gene expression on plant growth and fertility in Tx430, Tx623, and Tx2752, transgenic T0 plants transformed with pPHP79066 were sent to the greenhouse and grown to fertility. While subtle leaf curling occurred in Tx430, no severe pleiotropic effects were observed for all single copy T0 plants and all were fertile (Fig. 1m). There were, however, considerable pleiotropic effects observed for Tx623 and Tx2752 T0 plants (Fig. 1n, o). All the T0 Tx623 and Tx2752 plants were dwarf with strong curled leaves and reduced fertility, even for single-copy events. Because of the genotype-specific pleiotropic effects of morphogenic gene expression in sorghum, we decided to examine various transformation strategies that eliminate or preclude the morphogenic genes from the regenerating T0 plant.

**Morphogenic gene excision-induced selection-activation transformation system.** WUS is a bifunctional transcription factor that mediates stem cell homeostasis by regulating stem cell number and patterns of cell division and differentiation[10,17]. It acts as a transcriptional repressor in stem cell regulation and as a transcriptional activator in floral patterning[10]. In addition to causing pleiotropic effects on mature plant development as described above, it has been reported that morphogenic gene expression has a negative impact on transgenic plant regeneration through tissue culture[3,4,6]. It was reported that higher transformation efficiency was correlated with improved plant recovery after *Wus2/Bbm* cassette excision indicating the morphogenic genes may act differently at different tissue culture stages[4,5]. To maximize the positive, stimulatory effects of morphogenic genes on somatic embryo formation during tissue culture initiation, but to mitigate the repressive impacts on plant regeneration and rooting during the late tissue culture stage, an morphogenic gene excision-induced, selection-activation system was designed and successfully applied for maize transformation to generate high-quality, single-copy transgenic plants without pleiotropic

**Table 1 Transformation and event quality efficiencies of different sorghum transformation systems.**

| Transformation system | Construct | Selectable marker | Genotype | # of embryos infected | # of T0 plants (eff.)[a] | # of T0 QE (freq.)[b] | # of Marker-free T0 plants (eff.)[c] | # of T0 escape plants (freq.)[d] | # of Co-transformation in T0 plants (freq.)[e] |
|---|---|---|---|---|---|---|---|---|---|
| Conventional | pPHP86655 | NPTII | Tx430 | 472 | 99 (21.9%) | 58 (58.5%) | | 0 | |
| Integrated morphogenic gene-enabled | pPHP87980 | NPTII | Macia | 973 | 7 (0.7%) | NA | | NA | |
| | pPHP79066 ($Axig1_{pro}$:Wus2/$Pltp_{pro}$:Bbm) | Hra | Tx430 | 250 | 97 (38.8%) | 47 (48.5%) | | 0 | |
| | pPHP79066 ($Axig1_{pro}$:Wus2/$Pltp_{pro}$:Bbm) | Hra | Tx623 | 246 | 16 (6.5%) | NA | | NA | |
| | pPHP79066 ($Axig1_{pro}$:Wus2/$Pltp_{pro}$:Bbm) | Hra | Tx2752 | 200 | 19 (9.5%) | NA | | NA | |
| | pPHP81814 ($Axig1_{pro}$:Wus2/$Pltp_{pro}$:Bbm) | Hra | Tx430 | 248 | 173 (69.7%) | 66 (38.2%) | | 0 | |
| Morphogenic gene-enabled excision-induced selection-activation | pPHP81814 ($Axig1_{pro}$:Wus2/$Pltp_{pro}$:Bbm) | Hra | Macia | 249 | 50 (20.0%) | NA | | NA | |
| | pPHP81814 ($Axig1_{pro}$:Wus2/$Pltp_{pro}$:Bbm) | Hra | Tegemeo | 210 | 36 (17.1%) | NA | | NA | |
| | pPHP81814 ($Axig1_{pro}$:Wus2/$Pltp_{pro}$:Bbm) | Hra | Malisor 84-7 | 230 | 50 (21.7%) | NA | | NA | |
| | pPHP86482 ($Axig1_{pro}$:Wus2/$Pltp_{pro}$:Bbm) | Hra | Tx430 | 241 | 147 (60.9%) | 69 (46.9%) | | 0 | |
| Integrated *Wus2*-enabled | pPHP96564 ($Pltp_{pro}$:Wus2) | NPTII | Tx430 | 378 | 222 (58.7%) | 117 (52.7%) | | 0 | |
| | pPHP96564 ($Pltp_{pro}$:Wus2) | NPTII | Tx623 | 247 | 39 (15.8%) | 19 (48.7%) | | 0 | |
| *Wus2*/CRE-enabled marker-free | pPHP94632 ($Pltp_{pro}$:Wus2) | NPTII | Tx430 | 185 | 54 (29.2%) | 17 (31.4%) | 32 (59.2%) | 2 (3.7%) | |
| | pPHP94292 ($Pltp_{pro}$:Wus2) | NPTII | Tx430 | 169 | 54 (31.9%) | 19 (35.2%) | 33 (61.1%) | 0 | |
| Non-integrated *Wus2*-enabled altruistic | (90%) pPHP86655 + (10%) pPHP87078 (3xENH:$Pltp_{pro}$:Wus2/Sb-$Ubi_{pro}$:Zs-GREEN1) | NPTII | Tx430 | 370 | 135 (36.5%) | 76 (56.3%) | | 0 | 6 (4.4%) |
| | (90%) pPHP87980 + (10%) pPHP88158 (3xENH:$Pltp_{pro}$:Wus2/$NOS_{pro}$:Crc) | NPTII | Macia | 1072 | 162 (15.1%) | 95 (58.6%) | | 0 | 7 (4.3%) |

NA, not available.
[a]The number in the parentheses represents the transformation efficiency. The transformation efficiency was calculated as the number of regenerated shoots recovered per 100 embryos infected.
[b]For conventional and morphogenic gene-enabled transformations, the transgenic plant carrying a single copy of the intact T-DNA integration without vector backbone was defined as 'quality event (QE)'. For morphogenic gene-enabled excision-induced selection-activation transformation, the transgenic plant carrying a single copy of the intact T-DNA integration without vector backbone and Wus2/Bbm/moCRE/moCRE/GREEN1 cassette was defined as 'quality event (QE)'. For Wus2/moCRE-enabled marker-free transformation, the transgenic plants carrying a single copy of the intact T-DNA integration without vector backbone and Wus2/moCRE/NPTII cassette was defined as 'quality event (QE)'. For altruistic transformation, the transgenic plant carrying a single copy of the intact T-DNA integration without vector backbone and altruistic T-DNA co-integration was defined as 'quality event (QE)'. The number in the parentheses represents the percentage of QE.
[c]The number in the parentheses represents the percentage of Wus2/moCRE/NPTII cassette excision efficiency or marker-free efficiency.
[d]The number in the parentheses represents the percentage of non-transgenic escapes.
[e]The number in the parentheses represents the percentage of co-transformation.

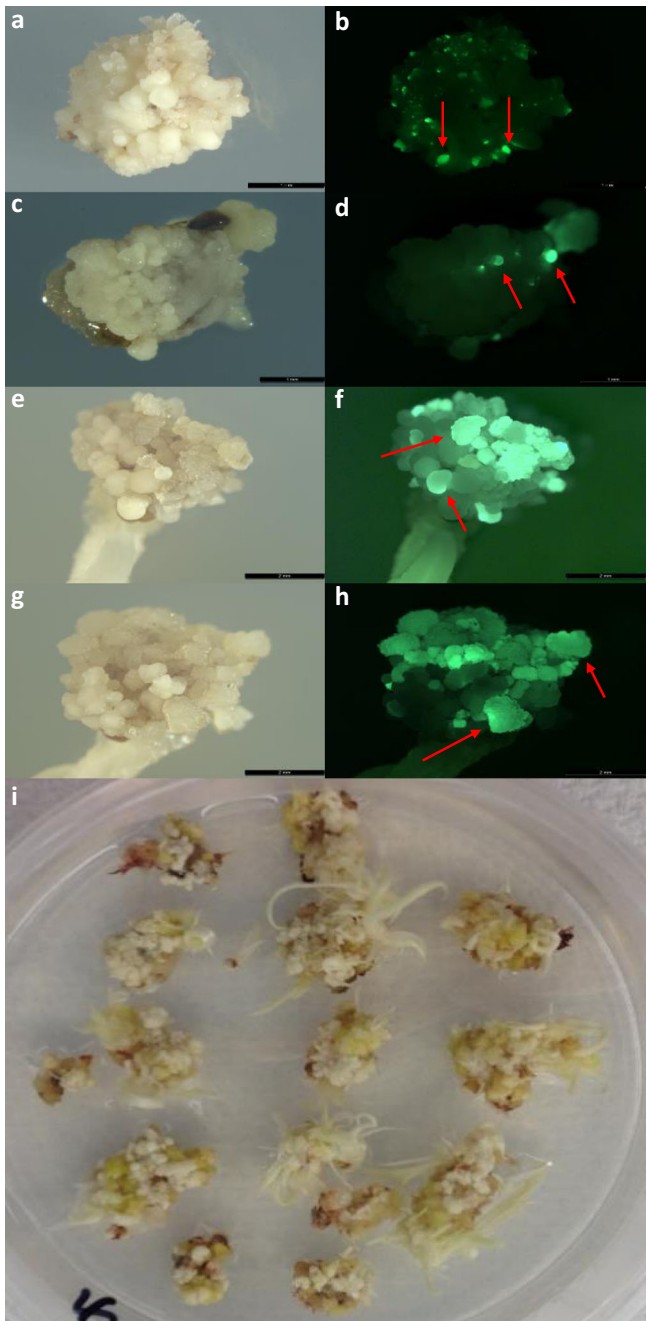

**Fig. 2 Somatic embryo formation and regeneration mediated by morphogenic gene excision-induced selection-activation system.** Bright field and fluorescence images of early-stage somatic embryo formation (couple of examples indicated by the arrows) on the surface of immature scutella in Tx430 (**a**, **b**), Macia (**c**, **d**), Malisor 84-7 (**e**, **f**), Tegemeo (**g**, **h**). **a**, **c**, **e**, and **g** Bright field images. **b**, **d**, **f**, and **h** Fluorescence images. **i** Tx430 shoot regeneration after four-week maturation.

impacts[18]. In brief, binary vector pPHP81814[12,18] (Supplementary Fig. 1b) was constructed with a multi-gene cassette containing *Wus2* (*Axig1_pro*:*Wus2*), *Bbm* (*Pltp_pro*:*Bbm*), *moCRE* (*Glb1_pro*:*moCRE*) and *Zs-GREEN1* (*Sb-Ubi_pro*:*Zs-GREEN1*) inserted into the first intron of the *Hra* gene, in which the *Wus2/Bbm/moCRE/GREEN1* cassette was flanked by two directly repeated *LoxP* sites. This design interrupts the *Hra* gene and ensures that the full length *Hra* gene cannot be expressed to confer herbicide resistance until the multi-gene cassette between

the two *LoxP* sites is removed. In other words, only those transgenic plants carrying at least one copy of the T-DNA with the *Wus2/Bbm/moCRE/GREEN1* cassettes effectively excised can be regenerated under selective pressure. Because the *Glb1* promoter driving *moCRE* is activated by ABA[19] in the maturation medium (Supplementary Table 5) through embryo development[12,18,20], *moCRE/loxP*-mediated *Wus2/Bbm/moCRE/GREEN1* cassette excision occurs during somatic embryo maturation, activating functional *Hra* expression before further regeneration[18].

To evaluate the transformation efficiency and single-copy quality event (QE) (see "Methods" for QE definition and determination) frequency using this recently designed excision-induced, selection-activation system, somatic embryo formation, and T0 plant regeneration were studied after inoculation with *Agrobacterium* carrying pPHP81814 for Tx430 and three Africa genotypes, namely Macia, Malisor 84-7 and Tegemeo, following the morphogenic gene-enabled transformation protocol (Supplementary Fig. 2). Robust somatic embryo formation, indicative of both morphogenesis and stimulation of cell division, was observed 14 days after *Agrobacterium* infection for all four genotypes as indicated by the globular-shaped protrusions from immature scutella (Fig. 2a–h). Those somatic embryos quickly germinated into plantlets as demonstrated in Tx430 (Fig. 2i) on maturation medium with selection (Supplementary Table 5). As shown in Table 1, the transformation efficiencies were as high as 69.7% for Tx430, 20.0% for Macia, 21.7% for Malisor 84-7, and 17.1% for Tegemeo, significantly higher than that of the conventional transformation previously reported[14]. The frequency of single copy insertion, backbone-minus, and *Wus2/Bbm/moCRE/GREEN1* cassette-free (QE%) was 38.2% for Tx430 transformed with pPHP81814. Using another production construct pPHP86482 derived from pPHP81814 (Supplementary Fig. 1b) containing a proprietary trait gene, the transformation efficiency in Tx430 was 60.9% of which 46.9% of the events were QEs (Table 1). No pleiotropic effects were observed for any of the pPHP86482-derived QE events sent to the greenhouse. In both examples described above for Tx430, the QE frequency of the excision-induced, selection-activation system (38.2–46.9%) was comparable to that of the integrated morphogenic gene-enabled transformation system (e.g., 48.5% QE frequency for pPHP79066) (Table 1). On the other hand, the transformation efficiency of the excision-induced, selection-activation system (60.9–69.7%) was significantly higher compared with the 38.8% transformation efficiency for pPHP79066, an integrated morphogenic gene-enabled transformation system (Table 1). This further supports the dual role of morphogenic genes for stimulating somatic embryo formation initially but inhibiting plant recovery and regeneration at the maturation stage. Therefore, higher transformation efficiency can be achieved by removing morphogenic genes after somatic embryo formation, but prior to germination.

**_Wus2/CRE_-enabled marker-free transformation system.** Most plant transformation technologies involve the random stable insertion of a transgene into the genome by coupling a trait gene with a selectable marker gene for antibiotic or herbicide resistance[21]. The presence of marker genes in transgenic plants, however, often provides no advantage after transformation and may raise regulatory concerns. It is desirable, therefore, to develop a highly efficient marker-free transformation technology for supporting the commercialization of genetically engineered crops[21].

It was reported that *Wus2* expression alone, driven by the maize *Pltp* promoter (hereby referred to as integrated *Wus2*-enabled transformation system that follows morphogenic

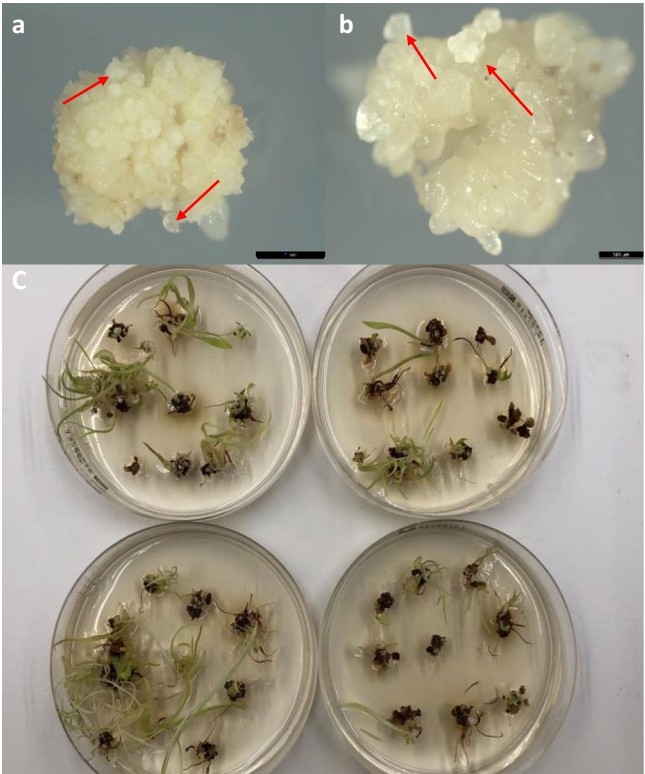

**Fig. 3 Somatic embryo formation and regeneration mediated by altruistic transformation system.** Bright field images of early-stage somatic embryo formation (couple of examples indicated by the arrows) on the surface of immature scutella induced by altruistic transformation in Tx430 (**a**) and Macia (**b**). **c** Macia shoot regeneration after four-week maturation following by one week in dim light.

gene-enabled transformation protocol as illustrated in Supplementary Fig. 2), was sufficient to rapidly stimulate somatic embryo formation and produce T0 maize plants[12]. As an example in sorghum using pPHP96564 (Supplementary Fig. 1g), an integrated Wus2-enabled transformation vector carrying $Pltp_{pro}$:Wus2 and $Zm\text{-}Ubi1_{pro}$:NPTII as the selectable marker, we demonstrated transformation efficiencies of 58.7 and 15.8% in Tx430 and Tx623, respectively, of which 52.7% of the Tx430 events and 48.7% of the Tx623 events produced were single copy QEs (Table 1). Corroborating the results initially reported in maize, these results clearly demonstrate the efficacy of $Pltp_{pro}$:Wus2 for stimulating somatic embryo formation in sorghum, in the absence of Bbm, and suggests that this may be genotype-independent. Taking advantage of this observation, it was recently reported that a heat shock-induced moCRE/LoxP-mediated excision system was developed to efficiently eliminate both Wus2 and selectable marker gene by heat-shock treatment before the maturation stage in maize[22] (hereby referred to as Wus2/CRE-enabled marker-free transformation system).

We tested this design in sorghum using binary vectors pPHP94632 and pPHP94292. As illustrated in Supplementary Fig. 1c and d, both binary vectors pPHP94632 and pPHP94292 contain Wus2 ($Pltp_{pro}$:Wus2), a heat shock inducible moCRE gene ($Hsp17.7_{pro}$:moCRE) and a selectable marker ($Zm\text{-}Ubi_{pro}$:NPTII) flanked by directly repeated loxP sites. pPHP94632 and pPHP94292 were identical plasmids, except for the proprietary trait gene expression cassette (Supplementary Fig. 1c, d). Following Agrobacterium infection and co-cultivation of Tx430 immature embryos, the immature embryos were then subcultured on multi-purpose medium with selection (250 mg l⁻¹

G418) for three weeks to induce somatic embryo formation (Supplementary Fig. 2). The intent of this tissue culture modification compared with the morphogenic gene-enabled transformation protocol (Supplementary Fig. 2) was to maximize transgenic somatic embryo formation, but at the same time minimize the proliferation of non-transgenic tissue so only the transgenic somatic embryos survived and germinated during maturation without selection. After the three-week-long stringent selection, excision of the Wus2/moCRE/NPTII cassette was carried out by inducing moCRE expression upon heat treatment at 45 °C and 70% humidity for 3 h (Supplementary Fig. 2). Following heat treatment, embryos were transferred to the maturation medium without selection for four weeks before rooting (Supplementary Fig. 2). As shown in Table 1, similar results were obtained for both binary vectors, pPHP94632 and pPHP94292. The three-week-long selection pressure during somatic embryo formation was stringent; except for two non-transgenic escapes (the escape frequency was between 0 and 3.7%), all 106 events from both experiments were transgenic. The heat shock-induced moCRE/LoxP-mediated excision system was highly efficient as well, with more than 59% of the events showing complete Wus2/moCRE/NPTII cassette excision (Table 1). Although the transformation efficiency was lower (in the range of 29.2–31.9%) compared to the excision-induced, selection-activation system described above (>60%), the overall QE frequency (single copy, backbone-free, and Wus2/moCRE/NPTII cassette-free) was in the range of 31.4–35.2% (Table 1). The high QE frequency of this marker-free system most likely resulted from the combined impact of highly efficient excision and extremely low non-transgenic escape frequency.

**Non-integrated Wus2-enabled altruistic transformation system.** WUS protein is non-cell-autonomous, migrating between cells via plasmodesmata[9,23]. It has been demonstrated that the movement of WUS protein stimulates somatic embryo formation of neighboring cells[11,12]. Taking advantage of this observation, a novel transformation system known as "altruistic" transformation, has been developed that utilizes the mixing of two Agrobacterium (Agro-1 and Agro-2), both being the same strain (LBA4404 Thy-) containing the accessory plasmid pPHP71539, but each Agrobacterium carrying one of two different plasmids for transformation[12]. Agro-1 carries an "altruistic" T-DNA binary plasmid with a Wus2 expression cassette but lacking a selectable marker gene, while Agro-2 carries a conventional binary T-DNA plasmid with a selectable marker and a gene-of-interest. Transformation is performed by mixing Agro-2:Agro-1 at a 9:1 ratio. During the initiation of transformation, the "altruistic" binary plasmid stimulates somatic embryogenesis in a non-cell-autonomous manner, and a morphogenic gene-enabled transformation protocol (Supplementary Fig. 2) can be followed for rapid and highly efficient transformation through somatic embryo formation from neighboring non-Wus2-containing cells. Using this method, the preponderance of regenerated plants contains the gene-of-interest T-DNA, but not the altruistic T-DNA containing Wus2 (hereby referred to as non-integrated Wus2-enabled altruistic transformation system).

To determine whether the altruistic method could be extended to sorghum, we compared the transformation efficiency between conventional vs. altruistic transformation in both Tx430 and Macia using two different altruistic T-DNA binary plasmids pPHP87078 and pPHP88158 (Supplementary Fig. 1e, f), respectively. Both altruistic T-DNA binary plasmids carried the same core component, $3xENH\text{:}Pltp_{pro}$:Wus2, to provide the "altruistic" stimulation, but with different visual marker genes. pPHP87078 carried $Ubi_{pro}$:Zs-GREEN1 as the visual marker gene

**Table 2 Targeted mutation and gene-dropout frequencies of different sorghum transformation systems.**

| Transformation system | Construct | sgRNA | Total # of T0 plants analyzed[a] | % of T0 mutation[b] freq. | % of T0 gene-dropout[c] freq. | # of T0 QEs | % of T0 QE mutation[b] freq. | % of T0 QE gene-dropout[c] freq. |
|---|---|---|---|---|---|---|---|---|
| Conventional | pPHP86655 | SB-Bmr6-CR4<br>SB-Bmr6-CR1 | 99 | 74.0 ± 8.5<br>76.2 ± 6.9 | 6.5 ± 2.9 | 58 | 70.8 ± 12.9<br>70.8 ± 12.9 | 3.1 ± 2.3 |
| Non-integrated Wus2-enabled altruistic | (90%) pPHP86655 + (10%) pPHP87078 (3xENH:Pltp_pro:Wus2/Sb-Ubi_pro:Zs-GREEN1) | SB-Bmr6-CR4<br>SB-Bmr6-CR1 | 135 | 66.8 ± 1.7<br>68.9 ± 1.5 | 20.5 ± 4.6 | 76 | 62.4 ± 4.2<br>65.8 ± 1.3 | 22.8 ± 6.5 |
| Integrated Wus2-enabled | pPHP96564 (Pltp_pro:Wus2) | SB-Bmr6-CR4<br>SB-Bmr6-CR1 | 222 | 93.7 ± 2.7<br>93.2 ± 2.0 | 43.9 ± 13.4 | 117 | 89.9 ± 4.0<br>89.9 ± 4.0 | 47.9 ± 18.4 |

Data were presented as the average ± SD of triplicated biological replications.
[a]The total T0 plants analyzed from three biological experiments.
[b]Mutation is defined as mutagenesis at target site, including both in-frame and frameshift mutations.
[c]Gene-dropout is defined as gene deletion between the two target sites.

and pPHP88158 carried $NOS_{pro}$:Crc, a gene-fusion between the maize *R* and *C1* genes that stimulates anthocyanin biosynthesis in planta[24] (Supplementary Fig. 1e, f). Similar to all other advanced transformation systems mediated by morphogenic genes, robust somatic embryo formation was observed for sorghum within 14 days on the multi-purpose medium without selection for both genotypes when the two *Agrobacterium* were mixed at the 9:1 ratio. This indicated that a mixture containing 10% altruistic *Agrobacterium* was sufficient to delivery *Wus2* T-DNA to the cells for inducing somatic embryo formation on the surface of immature scutella (Fig. 3a, b). As shown in Table 1, the baseline conventional transformation efficiency was about 21.9% for Tx430 and 0.7% for Macia using binary plasmids pPHP86655 and pPHP87980 (Supplementary Fig 1h, i), respectively. In contrast, by simply mixing the *Agrobacterium* carrying pPHP86655 or pPHP87980 with altruistic T-DNA binary plasmids pPHP87078 or pPHP88158 at a 9 to 1 ratio, respectively, the transformation efficiencies increased significantly to 36.5% for Tx430 and 15.1% for Macia (Table 1). Although 10% altruistic *Agrobacterium* (*Agro-1*) was sufficient to induce somatic embryo formation across the scutellum, the co-integration of both altruistic and conventional T-DNAs into one transgenic event was lower than 4.4% (Table 1). This resulted in a high QE frequency ranging from 56.3 to 58.6% (Table 1).

**WUS2 facilitates CRISPR/Cas-mediated high-frequency genome editing.** Targeted genome modification using the CRISPR/Cas system has proven to be a powerful tool for crop engineering and has been successfully applied to maize, rice, sorghum, and numerous other plant species to generate stable genome-edited events with targeted modifications (e.g., insertions, deletions, and replacements)[25–27]. An efficient transformation system is a prerequisite for CRISPR/Cas-mediated genome editing in plants[28]. Taking advantage of the highly efficient transformation mediated by morphogenic gene expression, we further conducted CRISPR/Cas-mediated genome editing and compared the targeted editing frequency to that of conventional transformation using the same guide-RNA(s) in sorghum. All the T0 plants were analyzed by deep sequencing to identify mutations at each target site and gene dropout between the two target sites ("Methods"). In the first experiment, sorghum ortholog of *Mtl*[29] (*Matrilinial Sobic.001G348600*) gene was mutated by targeting with a single guide RNA (sgRNA) (Supplementary Fig. 3). As shown in Supplementary Table 8, using the same sgRNA (*Sb-Mtl-CR1*), the mutation frequency in Tx430 increased 2.7-fold from 27.5% for conventional transformation using plasmid pPHP86801 (Supplementary Fig. 1j) compared to 74.2% when using the integrated morphogenic gene-enabled transformation plasmid pPHP87098 ($Axig1_{pro}$:Wus2/$Pltp_{pro}$:Bbm) (Supplementary Fig. 1k). In the second experiment, dropout a segment of the *Kaf* (γ-Kafirin, *Sobic.002G211700*) gene[30] was tested using two sgRNAs (*Sb-Kaf-CR1* and *-CR3*) (Supplementary Fig. 4) in Tx430. We compared both mutation frequency for each sgRNA and the gene-dropout frequency of those events generated by conventional transformation using plasmid PHP87018 (Supplementary Fig. 1l) and the non-integrated *Wus2*-enabled altruistic transformation (90% pPHP87018 + 10% altruistic vector pPHP88158 (Supplementary Fig. 1f)). We found that the mutation frequencies for each sgRNA, *Sb-Kaf-CR1* and *-CR3*, were increased from 39.3 and 48.5% for conventional transformation to 63.6% and 66.7% for non-integrated *Wus2*-enabled altruistic transformation, respectively (Supplementary Table 9). One gene-dropout event was identified using non-integrated *Wus2*-enabled altruistic transformation, but none were recovered from conventional transformation (Supplementary Table 9). The third experiment was

designed to delete the *Lgs1* (*Low germination stimulant 1, Sobic.005G213600*) gene[13,31] by gene dropout using two sgRNAs, *Sb-Lgs1-CR1* and *-CR3* (Supplementary Fig. 5), in the Macia background. Macia is barely transformable without morphogenic genes (0.7% transformation efficiency, as shown in Table 1). To generate enough edited events in Macia, we conducted integrated morphogenic gene-enabled transformation and non-integrated *Wus2*-enabled altruistic transformation and then compared mutation frequency for each sgRNA as well as the gene-dropout frequency. As shown in Supplementary Table 10, higher mutation frequencies of 83.3 and 92.9% at each target site and a higher gene-dropout frequency of 52.4% were observed when using integrated morphogenic gene-enabled transformation with pPHP87984 (Supplementary Fig. 1m) carrying *Axig1$_{pro}$:Wus2/ Pltp$_{pro}$:Bbm*. This compared to mutation frequencies of 40.4 and 57.8% and a 29.2% gene dropout frequency when using the non-integrated *Wus2*-enabled altruistic transformation method (90% pPHP87980 (Supplementary Fig. 1i) + 10% altruistic vector pPHP88158 (Supplementary Fig. 1f)).

Collectively, those results suggested a trend for improved gene editing frequency when utilizing *Wus2* during transformation. Namely, the highest frequency of genome editing was observed when using an integrated morphogenic gene-enabled transformation compared to that of non-integrated *Wus2*-enabled altruistic transformation, while the conventional transformation without *Wus*2 had the lowest gene-editing frequency. Assuming that plant cells harboring an integrated morphogenic gene accumulate more WUS2 protein than neighboring cells receiving WUS2 protein via migration from a cell expressing *Wus2*-enabled altruistic T-DNA, we hypothesize that CRISPR/Cas-mediated genome editing frequency may correlate with the level of WUS2 protein accumulated in the transgenic cells and that WUS2 facilitates more efficient genome editing.

The mutagenesis efficiency of the CRISPR/Cas9 system at the target site through the non-homologous end joining (NHEJ) chromosomal repair pathway is affected by multiple factors, including the expression level of *Cas9* and sgRNA, the sgRNA sequence and promoters driving both *Cas9* and sgRNA, etc.[32] To better control those variables and to eliminate the potential confounding effect of *Bbm* expression on CRISPR/Cas-mediated genome editing, we carefully redesigned the genome editing plasmid using the *Pltp* promoter driving *Wus*2 gene expression in the absence of *Bbm*. This allowed us to eliminate the spatial and temporal effects of different promoters on regulating *Wus*2 gene expression and WUS2 protein accumulation. As shown in Supplementary Fig. 6, same sgRNAs, *Sb-Bmr6-CR1* and *-CR4*, were designed for two new binary vectors, pPHP96564 (Supplementary Fig. 1g) and pPHP86655 (Supplementary Fig. 1h), to delete the *Bmr6* (*Brown midrib 6, Sobic.004G071000*) gene[33] in Tx430. pPHP86655 is a conventional transformation binary vector and pPHP96564 (*Pltp$_{pro}$:Wus2*) is an integrated *Wus2*-enabled transformation binary vector. To compare the targeted mutation frequency and gene-dropout frequency side by side between the different transformation systems, the immature sorghum embryos were isolated, mixed, and randomly split up into three groups for conducting conventional transformation using pPHP86655, non-integrated *Wus2*-enabled altruistic transformation using mixed two *Agrobacterium* with 90% pPHP86655 and 10% pPHP87078 (Supplementary Fig. 1e) and integrated *Wus2*-enabled transformation using pPHP96564 (*Pltp$_{pro}$:Wus2*). As shown in Table 2, the highest average mutation frequency of 93.7 ± 2.7% and 93.2 ± 2.0% at each target site of *Sb-Bmr6-CR1* and *-CR4*, respectively, was observed when utilizing integrated *Wus2*-enabled transformation, resulting in the highest average gene-dropout frequency of 43.9 ± 13.4%. The non-integrated *Wus2*-enabled transformation method resulted in a lower average mutation frequency

at each target site, 66.8 ± 1.7% and 68.9 ± 1.5% and a lower average 20.5 ± 4.6% gene-dropout frequency. In contrast, the conventional transformation method generated average mutation frequencies of 74.0 ± 8.5% and 76.2 ± 6.9% at each target site, similar to that of non-integrated *Wus2*-enabled altruistic transformation, but it had the lowest average gene-dropout frequency, 6.5 ± 2.9%, compared to either integrated *Wus2*-enabled transformation or non-integrated *Wus2*-enabled altruistic transformation. Once again, these results demonstrated a significant correlation of targeted genome editing frequency with the presumed level of WUS2 accumulation in the transformed cells.

The premise that Cas-mediated genome editing was stimulated during *Wus2*-mediated somatic embryogenesis was strengthened by being based on the edited data for four different genomic loci using seven total sgRNAs across four different chromosomes in two sorghum varieties, Tx430 and Macia. Though *Wus2* expression is sufficient to stimulate somatic embryo formation in tissue culture, the mechanism of how *Wus2* enhances CRISPR/Cas-targeted genome editing is not fully understood. One possibility could be that *Wus2*-enabled transformation may increase the chance of multi-copy T-DNA integration, in turn leading to enhanced Cas9 protein accumulation in the cells and resultant higher editing frequency. We excluded this possibility based on two observations. We first compared the QE frequencies (single-copy for T-DNA) among those transformation systems. As shown in Table 1, similar QE frequencies 58.5, 52.7, and 56.3% were observed for all three transformation systems using pPHP86655, 90% pPHP86655 plus 10% altruistic vector pPHP87078 and pPHP96564 (*Pltp:Wus2*), respectively, indicating that despite enhancing transformation efficiency, *Wus2*-enabled transformation had no impact on integrated T-DNA copy number in sorghum. We then compared the targeted genome editing frequency among the single-copy QEs exclusively to eliminate any potential issues related to multi-copy events. Consistent with above observation, the integrated *Wus2*-enabled transformation treatment once again showed the highest mutation frequency at both target sites (89.9 ± 4.0%) and gene-dropout frequency (47.9 ± 18.4%) when calculated for single-copy QE events alone (Table 2). This is followed by the non-integrated *Wus2*-enabled altruistic transformation treatment with an mutation frequency of 62.4 ± 4.2 and 65.8 ± 1.3% for each target site and a gene-dropout frequency of 22.8 ± 6.5% when calculated for the single-copy QE events (Table 2). Finally, conventional transformation produced mutation frequencies of 70.8 ± 12.9% at both target sites and the lowest gene-dropout frequency of 3.1 ± 2.3% when calculated for the single-copy QE events (Table 2). The consistency of these multiple lines of evidence described above strongly supports that *Wus2* expression enhances CRISPR/Cas-mediated genome editing in sorghum.

Cell division and dedifferentiation are two important processes for *de novo* shoot regeneration[8,34–36]. The activation of the homeodomain transcription factor WUS has been demonstrated to facilitate cell division and dedifferentiation to establish shoot stem cell niches during *de novo* shoot regeneration[8,37]. Compared to non-dividing cells that typically have more condensed DNA and stable nucleosomes, a constantly dividing cell rapidly replicates its DNA and undergoes constant chromatin reconfiguration during each cell cycle[38]. It is also known that dedifferentiation is regulated by various epigenetic mechanisms, such as histone acetylation which increases the accessibility of transcriptional regulators to target DNA and contributes to the regulation of gene expression[39]. Therefore, dedifferentiating plant cells possess more open, decondensed chromatin architecture[37,39,40]. Recently it is reported that WUS can also act via regulation of histone acetylation at its target loci in *Arabidopsis*[41], indicating that *WUS* activation may alter chromatin accessibility. Since CRISPR/Cas9-mediated editing

is more efficient in open chromatin regions than in closed chromatin regions[42–44], it is reasonable to speculate that WUS may, either directly or indirectly, increase chromatin accessibility thereby facilitating double-strand break (DSB) targeted by CRISPR/Cas among those cells stimulated to become somatic embryos during tissue culture. Higher DSB frequency leads to higher mutation efficiency at each target site and therefore, results in higher gene-drop out efficiency through the NHEJ chromosomal repair pathway. We did notice that although similar mutation efficiency was observed at each target site for both non-integrated *Wus2*-enabled altruistic transformation and conventional transformation, the gene-drop efficiency was significantly higher for non-integrated *Wus2*-enabled altruistic transformation as shown in Table 2. One explanation is that the gene-drop out process is more complex and requires coordinated and synchronized DSB at both target sites. The open chromatin in that region could provide a better environment for the gene fragment to be released once DSB on both targeted sites happens simultaneously. Otherwise, NHEJ repair may occur quickly and result in indel mutation at each target site preventing the gene fragment from being released. A carefully designed ATAC-seq assay may shed light on delineating the mechanism of how Wus2 enhances CRISPR/Cas-mediated genome editing in the future.

## Conclusions

Overall, in this study, we demonstrated that the *Wus2*-enabled transformation system is an integrated system for both highly efficient transformation and CRISPR/Cas-mediated genome editing in sorghum. Elucidating the mechanisms of how WUS2 interacts with CRISPR/Cas-mediated DSB outcomes will be important to improve homology-directed repair efficiency in the future and potentially be applied to other organisms, even beyond the kingdom plantae.

## Methods

***Agrobacterium* tumefaciens strain and plasmid vectors**. *Agrobacterium* auxotrophic strain LBA4404 Thy- carrying a ternary vector transformation system was used to generate transgenic sorghum plants and for CRISPR/Cas-mediated genome modifications. The ternary vector system contains the T-DNA binary vector and pVIR accessory plasmid (pPHP71539) as previously described by Che et al.[14] and Anand et al.[15] The T-DNA binary plasmid pPHP79066 (Supplementary Fig. 1a) carrying both *Wus2* and *Bbm* (*Axig1$_{pro}$*:*Wus2*/*Pltp$_{pro}$*:*Bbm*) as morphogenic genes, *Zs-YELLOW1* (*Ltp2$_{pro}$*:*Zs- YELLOW1*) as the visual marker and *Hra* (*Sb-Als$_{pro}$*:*Hra*) as the selectable marker was reported previously by Lowe et al.[3] The T-DNA binary plasmid pPHP81814 (Supplementary Fig. 1b) reported by Chu et al.[18] was a multi-gene cassette containing the *Wus2* (*Axig1$_{pro}$*:*Wus2*), *Bbm* (*Pltp$_{pro}$*:*Bbm*), *moCRE* (*Glb1$_{pro}$*:*moCRE*) and *Zs-GREEN1* (*Sb-Ubi $_{pro}$*:*Zs-GREEN1*) inserted in the first intron of *Hra* gene. The *Wus2*/*Bbm*/*moCRE*/*GREEN1* cassette was flanked by two directly repeated *LoxP* sites. Binary vectors pPHP94632 and pPHP94292 were used for marker-free transformation in which the excision cassette contains *Wus2* (*Pltp$_{pro}$*:*Wus2*), a heat shock inducible *moCRE* gene (*Hsp17.7$_{pro}$*:*moCRE*) and a selectable marker (*Zm-Ubi$_{pro}$*:*NPTII*) flanked by the directly repeated *lox*P sites (Supplementary Fig. 1c, d). Both marker-free T-DNA binary plasmids pPHP94632 and pPHP94292 were identical plasmids, except the same proprietary trait gene (Trait) was driven by different maize promoters (Supplementary Fig. 1c, d). Altruistic binary vectors pPHP87078 and pPHP88158 (Supplementary Fig. 1e, f) were reported previously by Hoerster et al.[12]

CRISPR/Cas gene modification was also achieved using the LBA4404 Thy- *Agrobacterium* strain carrying ternary vector transformation system described above. The SpCas9 and sgRNA gene modification machinery were expressed on a TDNA expressing binary vector, such as pPHP96564, pPHP86655, pPHP87980, pPHP86801, pPHP87098, pPHP87018, and pPHP87984 genome modification binary vectors described in this study (Supplementary Fig. 1g–m). The sgRNA target sites, sequences, and corresponding PAM for *Sb-Mtl*, *Sb-Kaf*, *Sb-Lgs1*, and *Sb-Bmr6* genes were illustrated in Supplementary Figs. 3–6.

The ternary design was assembled by first mobilizing the accessory plasmid pPHP71539 in the *Agrobacterium* auxotrophic strain LBA4404 Thy- and selecting on media supplemented with gentamycin (25 mg l$^{-1}$). Subsequently, the binary constructs were electroporated into *Agrobacterium* strain LBA4404 Thy- containing the accessory plasmid and recombinant colonies were selected on media supplemented with both gentamycin and spectinomycin. All constructs were then subjected to next-generation sequencing and sequence confirmation before conducting transformation experiments.

**Plant material, sorghum transformation, and transgenic event quality analysis**. Sorghum genotypes Tx430, Tx623, and Tx2752 and African genotypes (Macia, Malisor 84-7, and Tegemeo) grown in a greenhouse were used in this study. Tx430, Tx623 and Tx2752, Malisor *84-7* (PI 656048), and Macia (PI 565121) were requested from USDA GRIN. Tegemeo is a public line obtained from Mycogen Seeds. Immature embryo explants isolated from those sorghum plants were transformed with *Agrobacterium* auxotrophic strain LBA4404 Thy- a carrying a ternary vector transformation system to generate transgenic sorghum plants[14,15]. The conventional transformation method was performed as previously described by Wu et al.[16] and Che et al.[14] Morphogenic gene-mediated transformation was conducted by modifying the conventional sorghum transformation protocol, eliminating nine-weeks from the protracted selective callus proliferation phase as illustrated in Supplementary Fig. 2. The *Wus2*/*CRE*-mediated marker-free transformation tissue culture procedure is illustrated in Supplementary Fig. 2. As illustrated Supplementary Fig. 2, following the *Agrobacterium* infection and co-cultivation steps, the immature embryos were subcultured on the multi-purpose medium with selection for three weeks to induce somatic embryo formation. After three weeks of stringent selection, *Wus2*/*moCRE*/*NPTII* cassette excision was carried out by heat-shock treatment at 45 °C for 3 h (Supplementary Fig. 2). Following heat-shock treatment, embryos were transferred to the maturation medium without selection for four weeks before rooting (Supplementary Fig. 2). All the media described in this study for sorghum transformation were indicated in Supplementary Fig. 2 and attached in the supplementary tables (Supplementary Tables 1–6).

The integrated copy number of the T-DNA and the vector backbone in these transgenic plants were determined by a series of qPCR analyses[16,45]. For conventional and morphogenic gene-mediated transformations, the transgenic plant carrying a single copy of the intact T-DNA integrations without vector backbone was defined as 'quality event (QE)'. For morphogenic gene-mediated excision-induced selection-activation transformation, the transgenic plant carrying a single copy of the intact T-DNA integrations without vector backbone and with no *Wus2*/*Bbm*/*moCRE*/*GREEN1* sequences was defined as 'quality event (QE)'. For *Wus*/*CRE*-mediated marker-free transformation, the transgenic plant carrying a single copy of the intact T-DNA integrations without vector backbone and *Wus2*/*moCRE*/*NPTII* cassette was defined as 'quality event (QE)'. For altruistic transformation, the transgenic plant carrying a single copy of the intact T-DNA integrations without vector backbone and altruistic T-DNA co-integration was defined as 'quality event (QE)'.

**Microscopy and imaging**. Images were taken using a dissecting Leica M165 FC stereo-epifluorescence microscope, with YFP or GFP filters for detection of fluorescence, using the PLANAPO 1.0× objective, 0.63× zoom, and Leica Application Suite V4.7 acquisition software.

**CRISPR/Cas-mediated mutation frequency and gene-dropout frequency determination**. To detect mutations and gene-dropouts induced by the CRISPR/Cas9, T0 events generated through transformation were characterized by amplicon sequencing from DNA extracted from a single fresh leaf punch from each plant as per manufacturer recommendations via the sbeadexTM tissue extraction system (LGC Limited, UK). DNA was normalized to 10 ng/µl and twenty cycle target region PCR was performed on 50 ng of genomic DNA with PhusionTM Flash 2x master mix (Thermo Scientific, Waltham, MA) as per the manufacture's recommendations. Five microliters of primary PCR product were transferred to twenty cycles secondary amplification containing primers to attach individual sample indexes and sequencing components, again with PhusionTM Flash 2x master mix. Sequencing was performed on Illumina MiSeq®, paired-end 150 cycles per read according to Illumina standard operating procedure. Sequence reads were aligned to the wildtype reference sequence via Bowtie2. The allele return threshold was set to 5%. Mutation is defined as mutagenesis at target site, including both in-frame and frameshift mutations. Mutation frequency was calculated as the number of mutated events divided by the total number of events generated in a transformation experiment. Gene-dropout is defined as gene deletion between the two target sites. Gene-dropout frequency was calculated as the number of gene-drop events divided by the total number of events generated in a transformation experiment.

**Reporting summary**. Further information on research design is available in the Nature Research Reporting Summary linked to this article.

## Data availability

All relevant data supporting the findings of this work are available within the manuscript and the supporting materials. DNA sequences of amplicons related to Table 2 are available from the NCBI Sequence Read Archive under the BioProject no. PRJNA783007, accession no. SAMN23416143-SAMN23416598 (Supplementary Data 1).

## Materials availability

Novel biological materials described in this publication may be available to the academic community and other not-for-profit institutions solely for non-commercial research purposes upon acceptance and signing of a material transfer agreement between the author's institution and the requestor. In some cases, such materials may contain genetic elements described in the manuscript that were obtained from a third party(s) (e.g.,

*Zs-GREEN1*, *Zs-YELLOW1*, and *Cas9*) and the authors may not be able to provide materials including third party genetic elements to the requestor because of certain third-party contractual restrictions placed on the author's institution. In such cases, the requester will be required to obtain such materials directly from the third party. The author's and authors' institution do not make any express or implied permission(s) to the requester to make, use, sell, offer for sale, or import third party proprietary materials. Obtaining any such permission(s) will be the sole responsibility of the requestor. In order to protect Corteva Agriscience™ proprietary germplasm, such germplasm will not be made available except at the discretion of Corteva Agriscience™ and then only in accordance with all applicable governmental regulations.

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

## Acknowledgements

We thank the super binary vector construction team from Corteva Agriscience for their support with vector construction, genomic lab from Corteva Agriscience for deep-sequencing analysis, and PCR analysis characterization group from Corteva Agriscience for event quality analysis, and environmental control group from Corteva Agriscience for sorghum planting in the greenhouse. We thank Weiwei Zhu and Amanda Reed for performing sorghum transformation.

## Author contributions

P.C., E.W., M.S., M.A., and T.J. designed research and analyzed the data. P.C., E.W., M.S., M.A., W.G., and T.J. wrote the paper. A.A., K.L., H.G., A.S. involved in vector design. M.Y. performed PCR assays.

## Competing interests

The authors declare the following competing interests: All authors are or have been employees of Corteva Agriscience.

## Additional information

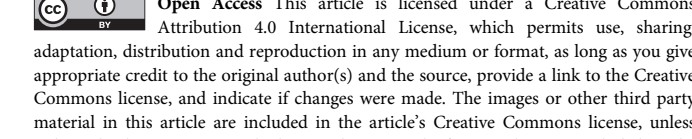

