## [Peer Review File · Communications Biology]

Reviewers' comments:

Reviewer #1 (Remarks to the Author):

To the Corresponding Author and Editors,

I read the Che et al manuscript with great interest. This body of work extends a number of technologies previously developed in maize (Hoerster et al 2020 and others) to sorghum to increase efficiency in recovering gene editing events through tissue culture.

This work validates those previously developed methods and widens the list of crop species known to efficiently respond to Wus2/Bbm expression for somatic embryo formation and tissue regeneration. In initial experiments, the choice to transform 3 sorghum genotypes (1 of high transformability and 2 recalcitrant lines) significantly strengthened the data and made a strong foundation for comparison.

It was interesting to read of the development of reduced methods that eliminated the nine-week selective callus proliferation phase described by Wu et al. (termed here morphogenic gene-enabled transformation system). Excitingly, the authors report timing from inoculation of immature embryos to plantlet transplantation at only two months (line 144) and an overall transformation efficiency at nearly 40% (line 146). These shortened methods should be of strong interest to the general plant transformation community.

The authors demonstrate a variety of expression cassette iterations for constitutive integration, developmentally and heat triggered excision and altruistic induction without integration of the morphogens genes. While these methods have been previously demonstrated in maize, the authors thoroughly investigate each approach in sorghum and their subsequent effects on transformation efficiency.

The authors conclude by investigating hypotheses centered on the positive effects of WUS on gene editing and gene dropout efficiencies. This theoretical analysis is well supported with the data at hand and the gene functions described in the literature.

The listed reagents/protocols in the supplementary will likely provide thorough details for reproduction by other researchers. Members of the plant community may benefit from a standardized method of obtaining vectors (such as Addgene submission) however it is understood that many if not all of the vectors listed are proprietary. Authors do not list agro concentrations (OD) in co-cultivation medium in the text (line 134; supplemental not available for reviewer download). Providing this information or citing other protocols that were reproduced may be helpful for readers to reproduce the method.

Overall I found the Chet et al. manuscript to have thorough experimentation, hypotheses and conclusions. This body of work will be of high interest to the plant transformation and genome editing community.

Best Regards,
Michael F. Maher PhD
Reviewer

Reviewer #2 (Remarks to the Author):

Previous research has demonstrated that co-expression of morphogenic genes such as Wus2 and BBM can promote shoot regeneration or meristem induction in crops and thus improve genetic transformation and engineering. In the present work, the authors successfully adapted systems including integrated morphogenic gene-enabled, marker-free and altruistic approaches for sorghum transformation and gene editing. The efficacy and efficiency have been greatly improved and the platform has the potential to overcome genotype dependency, compared with

conventional protocol. I have the following comments to be addressed.

1. The efficiencies of transformation systems may be not comparable where different sets of recalcitrant sorghum genotypes have been used. Please explain.
2. Table 1, why none of the recalcitrant African genotypes has OE events even all of them had a number of T0 plants regenerated?
3. A schematic process with a timeline for each system may be provided to avoid confusion.
4. Line 128, please give details about the promoters Axig1 and Pltp and their difference in driving capability.
5. Line 162, "this method also appears to overcome the genotype-dependent transformation barrier for previously non-transformable germplasms such as Tx623 and Tx2752" while in some inbred lines, the efficiency is as low as 1.4% so what's the possible reason?
6. Line 198, how to define "late embryo development"? Clear information is required here so that people can follow the protocol.
7. What is the non-transgenic escape frequency in the morphogenic gene-enabled excision-induced selection-activation system, given that the embryos are induced without selection pressure?
8. Line 319, co-integration of both altruistic and conventional T-DNAs into one transgenic event is very low so how is the WUS cassette functioning in the early stage of induction?
9. Line 360, the frequency of altruistic method is relatively low compared with integrated morphogenic gene-enabled transformation. Would the change of ratio i.e. 2:8 increase the efficiency than that of 1:9?
10. Line 429, "Wus2 expression enhances CRISPR/Cas-mediated genome editing in sorghum" but in fact, conventional transformation produced mutation frequencies of $70.8 \pm 12.9\%$ at both target sites, which is higher than Wus2-enabled altruistic transformation treatment with a mutation frequency of $62.4 \pm 4.2\%$ and $65.8 \pm 1.3\%$ for each target site. This result could not support the conclusion so please explain.
11. Wus2 expression enhances CRISPR/Cas-mediated mutation so would any other morphogenic genes such as BBM have a similar pattern? Please discuss.

Reviewer #3 (Remarks to the Author):

The key findings of the manuscript are the demonstration of a highly efficient CRISPR/Cas-targeted genome editing using Wus2-induced direct somatic embryo formation on a variety of Sorghum genotypes and rapid cycle time compared to conventional/traditional methods of transformation. The manuscript outlined a comprehensive study on a variety of Sorghum genotypes. It utilized various binary vector design strategies such as conventional, integrated morphogenic gene-enabled, morphogenic gene-enabled excision-induced selection-activation, marker-free, and altruistic to generate transgenic events with insertions of transgenes, marker-free events, edits, and gene dropouts. These events were further analyzed via detailed molecular analysis to demonstrate the robustness of the methodology, observations, and conclusions. In my view, this study represents a significant advancement in the genetic engineering of Sorghum. The overall theme presented in this manuscript is relevant and very valuable to the plant biology community engaged in research that utilizes genome-editing technologies for recalcitrant plant species.

Revisions

- 1) Change LAB4404 to LBA4404 (Line 468 and Line 507)
- 2) Figure 1: Replace the pictures represented in Figure 1 with better quality images (maybe close-up), especially Figure 1 d, e, f, j, k, and l.

Title: "*Wuschel2* enables highly efficient CRISPR/Cas-targeted genome editing during rapid *de novo* shoot regeneration in sorghum"

Reference No: COMMSBIO-21-2121-T

Authors: Che et al.

Dear Editor,

Thank you so much for the opportunity for publishing this manuscript. We are very pleased to receive considerable positive comments for the manuscript and appreciate all reviewers' recognition for the significant advance of the technology we developed for sorghum transformation and genome editing. We have revised the manuscript extensively as per the comments of the reviewers. In response to each of the reviewer's comments, we made the following revisions. All changes in the manuscript are highlighted in red.

Reviewer: #1

Remarks to the Author:

I read the Che et al manuscript with great interest. This body of work extends a number of technologies previously developed in maize (Hoerster et al 2020 and others) to sorghum to increase efficiency in recovering gene editing events through tissue culture.

This work validates those previously developed methods and widens the list of crop species known to efficiently respond to *Wus2/Bbm* expression for somatic embryo formation and tissue regeneration. In initial experiments, the choice to transform 3 sorghum genotypes (1 of high transformability and 2 recalcitrant lines) significantly strengthened the data and made a strong foundation for comparison.

It was interesting to read of the development of reduced methods that eliminated the nine-week selective callus proliferation phase described by Wu et al. (termed here morphogenic gene-enabled transformation system). Excitingly, the authors report timing from inoculation of immature embryos to plantlet transplantation at only two months (line 144) and an overall transformation efficiency at nearly 40% (line 146). These shortened methods should be of strong interest to the general plant transformation community.

The authors demonstrate a variety of expression cassette iterations for constitutive integration, developmentally and heat triggered excision and altruistic induction without integration of the morphogens genes. While these methods have been previously demonstrated in maize, the authors thoroughly investigate each approach in sorghum and their subsequent effects on transformation efficiency.

The authors conclude by investigating hypotheses centered on the positive effects of *WUS* on gene editing and gene dropout efficiencies. This theoretical analysis is well supported with the data at hand and the gene functions described in the literature.

The listed reagents/protocols in the supplementary will likely provide thorough details for reproduction by other researchers. Members of the plant community may benefit from a standardized method of obtaining vectors (such as Addgene submission) however it is understood that many if not all of the vectors listed are proprietary. Authors do not list agro concentrations (OD) in co-cultivation medium in

the text (line 134; supplemental not available for reviewer download). Providing this information or citing other protocols that were reproduced may be helpful for readers to reproduce the method.

Overall I found the Chet et al. manuscript to have thorough experimentation, hypotheses and conclusions. This body of work will be of high interest to the plant transformation and genome editing community.

Thanks for recognizing the value of the manuscript. OD value is provided in Supplementary Table 1.

Reviewer: 2

Remarks to the Author:

Previous research has demonstrated that co-expression of morphogenic genes such as Wus2 and BBM can promote shoot regeneration or meristem induction in crops and thus improve genetic transformation and engineering. In the present work, the authors successfully adapted systems including integrated morphogenic gene-enabled, marker-free and altruistic approaches for sorghum transformation and gene editing. The efficacy and efficiency have been greatly improved and the platform has the potential to overcome genotype dependency, compared with conventional protocol. I have the following comments to be addressed.

1. The efficiencies of transformation systems may be not comparable where different sets of recalcitrant sorghum genotypes have been used. Please explain.

We hope we understand this question correctly. As shown in Table 1, we compared transformation efficiency of all transformation systems established in Tx430 background and discussed the advantages and disadvantages for each system. Our intent was not to compare transformation efficiency among the genotypes with different transformation systems despite all being listed in the table 1. Rather, the purpose of listing the different genotypes is to illustrate that all genotypes tested could be transformed with reasonable transformation efficiency in the various systems tested. Table 1 simply provides an overview on how each system worked in different genotypes.

2. Table 1, why none of the recalcitrant African genotypes has QE events even all of them had a number of T0 plants regenerated?

The main reason is that transformation efficiency, but not event quality, is the essential parameter to answer if WUS2 can overcome the transformation barrier, particularly for genome editing purposes. We started with three sorghum genotypes (one most transformable Tx430 and two non-transformable 623 and 2752 genotypes) to address what causes genotype-dependent conventional transformation and whether WUS2 can overcome the genotype-dependent transformation barrier for previously non-transformable germplasms. Therefore, extensive studies were carried out to determine both transformation efficiency and event quality for all the transformation systems in Tx430. Once the transformation systems were established, we extended the study to African varieties, especially for Macia which is barely transformable using conventional method. In fact, both transformation efficiency and QE were determined for Macia for one of the experiments as shown in Table 1 and the QE frequency was consistent with Tx430. We did not, however, identify QE events for all the experiments related to Macia and other African varieties to save resources and to simply focus on one question "Can

WUS2 overcome the genotype-dependent transformation barrier for Macia based transformation efficiency determined?”.

3. A schematic process with a timeline for each system may be provided to avoid confusion.

Timelines are added to the flow diagram in Supplementary Figure 2. Each transformation system and the timelines are color coded.

4. Line 128, please give details about the promoters *Axig1* and *Pltp* and their difference in driving capability.

Plasmid pPHP79066 carrying both *Axig1* and *Pltp* promoters was previously tested by Lowe et al. 2018 in maize and the promoter details regarding to the features of *Axig1* and *Pltp* were well addressed/discussed in the publication. In brief, *Axig1* is an auxin-inducible promoter and *PLTP* is developmentally regulated, being specifically expressed in the scutellum epithelium tissue during embryo development. The publication by Lowe et al. was previously cited as citation 3 in line 129. To better address review’s concerns, citation was added to line 127 as well, directly link to the plasmid pPHP79066. The phrase “previously tested in maize” was added to line 127 for clarity.

5. Line 162, “this method also appears to overcome the genotype-dependent transformation barrier for previously non-transformable germplasms such as Tx623 and Tx2752” while in some inbred lines, the efficiency is as low as 1.4% so what’s the possible reason?

Although *Wus2* facilitates transformation by overcoming the genotype-dependent transformation barrier for previously non-transformable germplasms, transformation efficiency varies among the genotypes. One possible reason could be that different genotypes response to the tissue culture media differently which could result in different transformation and regeneration efficiency. Detailed studies are required to address this question in the future. Nevertheless, the main point we are trying to address here is if all genotypes we tested are transformable regardless of transformation efficiency.

6. Line 198, how to define “late embryo development”? Clear information is required here so that people can follow the protocol.

Glb1 promoter expression during embryo development was well characterized by Belanger and Kriz, 1989 (citation19). Regulation *Glb1* expression by ABA was reported by Liu et al. 1998 (citation20). Vector pPHP81814 carrying *Glb1_{pro}:moCRE* was previously tested by chu et al. 2019 (citation18) and Hoerster et al. 2020 (citation17) in maize. To clarify the concerns from the reviewer and to make the statement more accurate, we revised the phrase “late embryo development” in the sentence in line 199 to the following.

“Because the *Glb1* promoter driving *moCRE* is activated by ABA¹⁹ in the maturation medium (Supplementary Table 5) through embryo development^{12, 18, 20}, *moCRE/loxP*-mediated *Wus2/Bbm/moCRE/GREEN1* cassette excision occurs during somatic embryo maturation, activating functional *Hra* expression before further regeneration¹⁸”.

All citations described above, 12, 18, 19 and 20, were added to line 199 and 200, respectively. Citations 12 and 18 are added to line 191 for vector pPHP81814 as well.

7. What is the non-transgenic escape frequency in the morphogenic gene-enabled excision-induced selection-activation system, given that the embryos are induced without selection pressure?

Although embryogenesis was induced before selection pressure was applied, only those embryos carrying at least one copy of the T-DNA with the *Wus2*/*Bbm*/moCRE/GREEN1 cassettes effectively excised were able to restore *Hra* gene expression and gain selection pressure during maturation with selection later. Because of the stringent selection with imazapyr during maturation, the non-transgenic escape frequency was 0% for the morphogenic gene-enabled excision-induced selection-activation system. To address reviewer's concern, now we added escape frequencies for all the experiments with event quality analyzed to Table 1.

8. Line 319, co-integration of both altruistic and conventional T-DNAs into one transgenic event is very low so how is the WUS cassette functioning in the early stage of induction?

Because WUS protein is non-cell autonomous, migrating between cells via plasmodesmata. Co-integration was not required for WUS2 to stimulate embryogenesis around the neighboring cells. Despite lower level of altruistic agro applied for transformation, WUS protein can still stimulate somatic embryo formation not only for the few cells with integrated altruistic T-DNA, but also for many neighboring cells without carrying altruistic T-DNA insertion. That's why we used only one tenth of agro carrying altruistic T-DNA plasmid to avoid co-integration with conventional binary T-DNA so that more quality events without carrying WUS2 can be recovered. We believe that the function of altruistic WUS cassette was well described/discussed from line 284-300 in the manuscript.

9. Line 360, the frequency of altruistic method is relatively low compared with integrated morphogenic gene-enabled transformation. Would the change of ratio i.e. 2:8 increase the efficiency than that of 1:9?

This is a good question. The altruistic transformation conducted in sorghum was based on what we learned in maize as described by Hoerster et al. 2020 (citation17). We were trying to demonstrate that it's an alternative but efficient method for sorghum transformation to quickly generate events without *Wus2* integration. Altruistic method was well established in maize and 1:9 ratio was demonstrated to be the optimal ratio to achieve high transformation efficiency along with decent QE efficiency for most genotypes. Although higher ratio may increase transformation efficiency further, it could also reduce the QE frequency because of the increased chance for T-DNA co-integration. However, for some recalcitrant sorghum varieties with very low transformation efficiency, it is worth to test different ratios to improve the transformation first before considering the event quality efficiency.

10. Line 429, "Wus2 expression enhances CRISPR/Cas-mediated genome editing in sorghum" but in fact, conventional transformation produced mutation frequencies of 70.8±12.9% at both target sites, which is higher than Wus2-enabled altruistic transformation treatment with a mutation frequency of 62.4±4.2% and 65.8±1.3% for each target site. This result could not support the conclusion so please explain.

CRISPR/Cas-mediated mutagenesis efficiency and gene dropout efficiency are affected by many factors, such as the efficiency of gRNAs, chromatin architecture at different region of the genome, so on so forth and many other unknown factors. When WUS protein accumulation at moderate level, such as the case in altruistic transformation, the effect could be subtle, case by case, and maybe dominated/influenced by other factors to make the WUS2 effect less obvious. Therefore, we draw the overall conclusion based on comprehensive analysis of multiple lines of evidence by comparing edited data of three transformation systems (conventional, altruistic and integrated *Wus2*-enabled) for

four different genomic loci using seven total sgRNAs in both Tx430 and Macia backgrounds. In all cases, the integrated *Wus2*-enabled transformation system with the highest level of WUS2 protein accumulation showed the highest mutation frequency at all target sites and resulted in the highest gene dropout efficiency as well. On the contrary, conventional transformation with no WUS2 protein accumulation was always the lowest for both mutagenesis efficiency at each target site and gene dropout efficiency. To further support this notion, we further performed genome editing with altruistic transformation in which WUS2 protein accumulation relies on protein migration to the neighboring cells carrying the editing components, but without WUS2 gene integration and then compared to both conventional and integrated *Wus2*-enabled transformation systems as described in the manuscript. We prefer not to say that $70.8 \pm 12.9\%$ is significantly higher than $62.4 \pm 4.2\%$ and $65.8 \pm 1.3\%$ due to the experimental variations. Considering the standard error, those numbers are statistically non-distinguishable. Although it was not distinguishable for mutation frequency for this specific experiment, the gene dropout efficiency was significantly higher for altruistic ($22.5 \pm 6.5\%$) compared to that of conventional ($3.1 \pm 2.3\%$). The possible reasons were discussed in depth from line 450 to 460 in the manuscript. In a different experiment, however, as shown in Supplementary Table 9, the mutation frequency for altruistic (higher than 63.6%) was significantly higher than the conventional (lower than 48.5%) and higher gene dropout efficiency was also observed which, once again, supports the overall conclusion.

11. *Wus2* expression enhances CRISPR/Cas-mediated mutation so would any other morphogenic genes such as BBM have a similar pattern? Please discuss.

This is a good question. We are currently working on demonstrating if BBM alone has the similar pattern on genome editing or if additional effect can be observed when working with WUS2 together. This could be an interesting topic for another manuscript. For this manuscript, however, we chose to focus on WUS2 only to demonstrate that WUS2 alone without BBM is sufficient to overcome the genotype-dependent transformation barrier and enhance genome editing.

Reviewer: 3

Remarks to the Author:

The key findings of the manuscript are the demonstration of a pattern efficient CRISPR/Cas-targeted genome editing using *Wus2*-induced direct somatic embryo formation on a variety of Sorghum genotypes and rapid cycle time compared to conventional/traditional methods of transformation. The manuscript outlined a comprehensive study on a variety of Sorghum genotypes. It utilized various binary vector design strategies such as conventional, integrated morphogenic gene-enabled, morphogenic gene-enabled excision-induced selection-activation, marker-free, and altruistic to generate transgenic events with insertions of transgenes, marker-free events, edits, and gene dropouts. These events were further analyzed via detailed molecular analysis to demonstrate the robustness of the methodology, observations, and conclusions.

In my view, this study represents a significant advancement in the genetic engineering of Sorghum. The overall theme presented in this manuscript is relevant and very valuable to the plant biology community engaged in research that utilizes genome-editing technologies for recalcitrant plant species.

Revisions

1) Change LAB4404 to LBA4404 (Line 468 and Line 507):

This is a good catch. Both were corrected accordingly in Line 470 and Line 509.

2) Figure 1: Replace the pictures represented in Figure 1 with better quality images (maybe close-up), especially Figure 1 d, e, f, j, k, and l.

We rotated Figure 1 from vertical to horizontal to make the figure bigger and increase the size of all pictures. We replaced Fig1 d and j with new pictures to better demonstrate the robust callus and shoot development for Tx430. We did not provide close-up look and mainly want to demonstrate how the population of embryos, instead of selected individual ones, responded to the tissue culture (Figure 1 d, e, f) without *Wus2* and regenerate to form shoots with *Wus2* (Figure 1 j, k, and l). In addition, we also improved the image quality for Figure 1a, b, c, g, h and i.

REVIEWERS' COMMENTS:

Reviewer #1 (Remarks to the Author):

Reiteration of original manuscript review:

I read the Che et al manuscript with great interest. This body of work extends a number of technologies previously developed in maize (Hoerster et al 2020 and others) to sorghum to increase efficiency in recovering gene editing events through tissue culture.

This work validates those previously developed methods and widens the list of crop species known to efficiently respond to Wus2/Bbm expression for somatic embryo formation and tissue regeneration. In initial experiments, the choice to transform 3 sorghum genotypes (1 of high transformability and 2 recalcitrant lines) significantly strengthened the data and made a strong foundation for comparison.

It was interesting to read of the development of reduced methods that eliminated the nine-week selective callus proliferation phase described by Wu et al. (termed here morphogenic gene-enabled transformation system). Excitingly, the authors report timing from inoculation of immature embryos to plantlet transplantation at only two months (line 144) and an overall transformation efficiency at nearly 40% (line 146). These shortened methods should be of strong interest to the general plant transformation community.

The authors demonstrate a variety of expression cassette iterations for constitutive integration, developmentally and heat triggered excision and altruistic induction without integration of the morphogens genes. While these methods have been previously demonstrated in maize, the authors thoroughly investigate each approach in sorghum and their subsequent effects on transformation efficiency.

The authors conclude by investigating hypotheses centered on the positive effects of WUS on gene editing and gene dropout efficiencies. This theoretical analysis is well supported with the data at hand and the gene functions described in the literature.

In response to questions the authors addressed concerns and supplemental was added online for review.

Overall I found the Chet et al. manuscript to have thorough experimentation, hypotheses and conclusions. This body of work will be of high interest to the plant transformation and genome editing community.

Best Regards,
Michael Maher

Reviewer #2 (Remarks to the Author):

the authors have properly addressed my previous comments and suggestions. I would recommend accepting the paper